# L amino acid transporter structure and molecular bases for the asymmetry of substrate interaction

Ekaitz Errasti-Murugarren [1,12], Joana Fort [1,2,3,12], Paola Bartoccioni[1,2,12], Lucía Díaz[4], Els Pardon[5,6], Xavier Carpena[7], Meritxell Espino-Guarch [8], Antonio Zorzano [1,3,9], Christine Ziegler[10], Jan Steyaert [5,6], Juan Fernández-Recio[4], Ignacio Fita[11] & Manuel Palacín[1,2,3]

L-amino acid transporters (LATs) play key roles in human physiology and are implicated in several human pathologies. LATs are asymmetric amino acid exchangers where the low apparent affinity cytoplasmic side controls the exchange of substrates with high apparent affinity on the extracellular side. Here, we report the crystal structures of an LAT, the bacterial alanine-serine-cysteine exchanger (BasC), in a non-occluded inward-facing conformation in both apo and substrate-bound states. We crystallized BasC in complex with a nanobody, which blocks the transporter from the intracellular side, thus unveiling the sidedness of the substrate interaction of BasC. Two conserved residues in human LATs, Tyr 236 and Lys 154, are located in equivalent positions to the Na1 and Na2 sites of sodium-dependent APC superfamily transporters. Functional studies and molecular dynamics (MD) calculations reveal that these residues are key for the asymmetric substrate interaction of BasC and in the homologous human transporter Asc-1.

[1] Institute for Research in Biomedicine (IRB Barcelona), Barcelona Institute of Science and Technology, 08028 Barcelona, Spain. [2] Centro de Investigación Biomédica en Red de Enfermedades Raras (CIBERER), 08028 Barcelona, Spain. [3] Department of Biochemistry and Molecular Biomedicine, Faculty of Biology, University of Barcelona, 08028 Barcelona, Spain. [4] Barcelona Supercomputing Center (BSC), Joint BSC-CRG-IRB Research Program in Computational Biology, Life Sciences Department, 08034 Barcelona, Spain. [5] VIB-VUB Center for Structural Biology, Pleinlaan 2, 1050 Brussels, Belgium. [6] Structural Biology Brussels, Vrije Universiteit Brussel, Pleinlaan 2, 1050 Brussels, Belgium. [7] CELLS-ALBA Synchrotron Light Source, 08290 Barcelona, Spain. [8] Translational Medicine, Sidra Medicine, 26999 Doha, Qatar. [9] Centro de Investigación Biomédica en Red de Diabetes y Enfermedades Metabólicas Asociadas (CIBERDEM), 08028 Barcelona, Spain. [10] Institute of Biophysics and Biophysical Chemistry, Universität Regensburg, 95053 Regensburg, Germany. [11] Barcelona Molecular Biology Institut (IBMB-CSIC) and Unit of Excellence María de Maeztu, 08028 Barcelona, Spain. [12] These authors contributed equally: Ekaitz Errasti-Murugarren, Joana Fort, Paola Bartoccioni. Correspondence and requests for materials should be addressed to I.F. (email: ifrcri@ibmb.csic.es) or to M.P. (email: manuel.palacin@irbbarcelona.org)

Amino acid availability regulates cellular physiology. Specialized amino acid transporters mediate the transfer of amino acids across the plasma membrane. In humans, ten types of transporters are known to be responsible for the traffic and balance of amino acids within and between cells and tissues. Of these, the heteromeric amino acid transporters (HATs) form protein complexes composed of two polypeptides: a heavy or ancillary subunit and a light or transporter subunit. The heavy subunits (SLC3 family, rBAT, and CD98hc) first appeared in metazoans and form disulfide-linked heterodimers with the catalytic transporter subunits of one of the two SLC7 subfamilies, the L-type amino acid transporters (LATs)[1]. LATs are obligatory exchangers, which are also found in bacteria (e.g., SteT, BasC)[2,3] and transport all common amino acids except proline[1].

As obligatory exchangers, LATs do not modify the overall amino acid concentration gradient between the two sides of the cell membrane, but rather act on the proportional distribution of particular amino acids, thus regulating intracellular amino acid pools[4]. The asymmetric functional interaction of LATs with the substrates at both sides of the plasma membrane thus allows the high concentration of intracellular substrates (mM range) to control the exchange with substrates in the extracellular medium (μM range)[5].

The polar distribution of some HATs in the apical and basolateral membranes of polarized epithelia, such as renal tubule and small intestine, regulates amino acid (re)absorption and whole body amino acid homeostasis[6–8]. Consistent with the key role of these transporters in metabolism and physiology, dysfunctional HATs cause human disease. For example, mutations in b[0,+]AT and y[+]LAT1 transporters, which participate in the renal reabsorption of amino acids, cause primary inherited aminoacidurias (e.g., cystinuria and lysinuric protein intolerance)[8,9]. Moreover, characterization of knockout mouse models of LATs and further sequencing of the human genome have enabled the identification of underlying causative mutations and revealed new roles for LAT subfamily members in the function of multiple organs. Thus, mutations in LAT1 cause autism spectrum disease[10], while those in LAT2 contribute to age-related hearing loss[11]. Additionally, in the brain, Asc-1 is the major transporter of D-serine, which acts as a co-agonist of NMDA-glutamate receptors and is thus a novel target to treat schizophrenia[12]. Finally, LAT1 and xCT are potential targets for cancer therapy because they are over-expressed in many tumors[13,14].

Despite the relevance of HATs, knowledge of their structure is limited to the atomic structure of the extracellular domain of the heavy subunit CD98hc (SLC3A2)[15] and to a low-resolution model of human LAT2/CD98hc[16]. Additionally, the crystal structures of the pH-activated sodium-independent amino acid facilitator ApcT and the amino acid transporters AdiC and GadC, distant bacterial homologs of human LATs, which belong to the amino acids, polyamines and organoCations (APC) family of transporters, have been solved[17–22]. Furthermore, the crystal structure of GkApcT, a homolog of the cationic amino acid transporter (CAT) subfamily, the other subfamily of the SLC7 transporters, has been recently solved[23]. All of these bacterial transporters belong to the APC family, within the APC superfamily, and adopt the APC superfamily-fold, originally described for the bacterial transporter LeuT[24]. Structural models based on AdiC and ApcT have been used to obtain clues regarding substrate recognition by LAT1[25] and LAT2[16,26]. However, due to the low sequence identity with human LATs (14–22%) these models lack the robustness required to dissect the molecular mechanisms underlying transport, study the molecular defects associated with human mutations, and build structure-guided inhibitors with potential pharmacological applications.

To fill this gap in our knowledge, we sought to determine the crystal structures of a prokaryotic LAT transporter, the bacterial alanine-serine-cysteine exchanger (BasC), which has 26–28% sequence identity with human LATs[3]. BasC crystallizes in complex with a nanobody (Nb), in a non-occluded inward-facing conformation in the apo state, and in complex with the substrate analog 2-aminoisobutyrate (2-AIB). Also, we exploit the structural homology of APC superfamily transporters and the known pathological missense mutations of human LATs to identify key residues involved in transporter function. Our findings reveal the first determinants of the asymmetry of the apparent substrate affinity at the two sides of the plasma membrane in LATs.

## Results

**BasC structure.** BasC was co-crystallized with a newly generated anti-BasC nanobody (Nb74) in the absence and presence of the amino acid analog 2-AIB, and the structures were determined to 2.9 and 3.4 Å resolution, respectively (Fig. 1). The BasC-Nb74 complex used for crystallization assays was purified by SEC (Supplementary Fig. 1c). Data collection and refinement statistics are summarized in Supplementary Table 1. The apo structure (without 2-AIB) was solved by molecular replacement and confirmed with data at 4.3 Å resolution from BasC (SeMet)-Nb74 crystals (Supplementary Table 1 and Supplementary Fig. 2a). Crystals of both the apo and holo BasC-Nb74 complex are in space group P4$_1$2$_1$2, and they contain a single copy of the complex in their asymmetric unit and identical crystal packing. Supplementary Fig. 1a shows the packing of the apo BasC-Nb74 complex, with the Nb having a key role in structural packing interactions.

BasC contains 12 transmembrane (TM) helices with the N-termini and C-termini located intracellularly (Fig. 1a, b), according to the membrane topology of human LATs[27]. In addition, the unambiguous alignment with human LATs allowed us to locate all the reported missense mutations in the structure of BasC causing disease (Supplementary Fig. 2b). BasC adopts the APC superfamily fold[28]. Thus, TM1–TM5 and TM6–TM10 are related by a pseudo two-fold symmetry axis within the plane of the membrane. Both TM1 and TM6 are unwound in the center, forming two discontinuous helices named 1a, 1b, and 6a, 6b (Fig. 1a, b). Two domains can be distinguished in the BasC structure, namely the bundle comprised by TM1, 2, 6, and 7, and the hash domain formed by TM3, 4, 8, and 9, with TM5 and TM10 connecting the two domains at each side of the transporter (Fig. 1a and Supplementary Fig. 3). Finally, TM11 and 12 form a V-shape at the external side of TM10 (Fig. 1b). The apo and holo structures are in an inward (cytoplasmic)-facing non-occluded conformation with TM1a and TM6b tilted to open a vestibule connecting the center of the transporter with the cytoplasm (Fig. 1a–c). By contrast, the central vestibule is not connected to the extracellular space (Fig. 1c, d). Indeed, hydrophobic residues in TM1b (Phe 24), TM3 (Ile 105), TM6a (Phe 199) and TM10 (Phe 343), and polar residues in TM1b (Lys 25), extracellular loop 4 (Asp 256), and TM7 (Asn 344 and Thr 347) build a thick external gate (Fig. 1d).

**Structural and functional interaction between BasC and Nb74.** Nb74 interacts at the cytoplasmic face of BasC (Fig. 1a) with high affinity (3.9 ± 1.4 nM) (Supplementary Fig. 1b) and with 1:1 stoichiometry (Fig. 1a and Supplementary Fig. 1a). BasC not bound to Nb74 was also crystallized and solved at low resolution (7.2 Å) in inward-facing state (Supplementary Fig. 3), thereby suggesting that Nb74 does not sample the transporter in inward-facing conformation. The BasC-Nb74 interaction interface is formed principally by the complementary determining regions 1 and 3 of

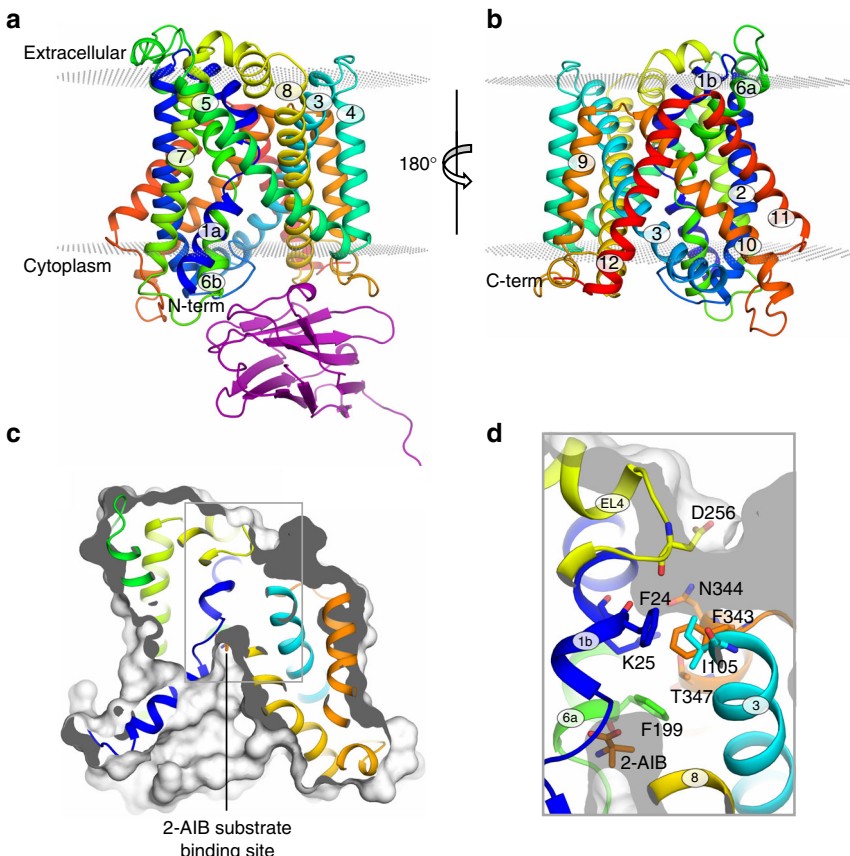

**Fig. 1** Crystal structures of BasC. **a**, **b** Crystal structure of BasC apo in complex with nanobody 74 (Nb74) at 2.9 Å (PDB ID 6F2G). Helices are colored blue to red from the N-termini and Nb74 is shown in magenta. **c**, **d** Crystal structure of BasC-Nb74 in complex with 2-AIB at 3.4 Å (PDB ID 6F2W). Helices are colored as in **a**, **b**, and the protein surface is depicted in light gray. The sagittal plane shows the substrate 2-AIB (orange) at the end of the vestibule open to the cytoplasm. **d** Detail of region squared in **c** showing the residues in the thick barrier that prevents the substrate from accessing the extracellular medium

**Table 1 Kinetic parameters for ʟ-alanine in BasC wild-type and mutants**

|  | $K_m$ (o) (μM) | $K_m$ (i) (μM) | $V_{max}$ (pmols ʟ-alanine μg protein$^{-1}$ s$^{-1}$) |
|---|---|---|---|
| Wild-type | 53.73 ± 3.97 | 2220 ± 150 | 6.37 ± 0.70 |
| Y236F | 50.25 ± 3.44 | 560 ± 110*** | 4.58 ± 0.17 |
| K154A | 474 ± 42*** | 2040 ± 370 | 0.32 ± 0.04*** |

Student's t-test ***$p < 0.001$, Wild-type values vs K154A Km (o), Y236F Km (i) and K154A $V_{max}$

the Nb and the BasC intracellular side of TM6b, 8, and 9 and intracellular loops 1 and 4, where mainly polar and electrostatic interactions are present (Supplementary Fig. 4a). Nb74 recognizes residues from TM6b in bundle domain and TM8 and TM9 in hash domain (Supplementary Fig. 4a), thereby pointing to a possible role of Nb74 in the inhibition of BasC activity.

The apparent substrate affinity of BasC differs at each side of the membrane (e.g., ~50 μM and ~2000 μM for ʟ-alanine for the extracellular and intracellular sites, respectively) (see ref. [3] and Table 1). Nevertheless, insertion of BasC in PLs occurs randomly[3] and addition of Nb74 to the medium blocked only inside-out oriented BasC molecules (i.e., BasC molecules with the cytoplasmic side facing the PL external medium) (Fig. 2a). Indeed, addition of Nb74 to the PL medium significantly blocked 10 μM [³H]ʟ-alanine efflux (IC₅₀ of ~300 nM Nb74; Supplementary Fig. 4b) without affecting [³H]ʟ-alanine influx (Fig. 2a). This observation indicates that 10 μM [³H]ʟ-alanine transport by inside-out inserted BasC transporters is negligible. Indeed, the

sidedness of the apparent affinities of the transporter for substrate can be evidenced by using Nb74 in the transport medium to block [³H]ʟ-alanine uptake mediated by the inside-out inserted BasC molecules (Fig. 2b). Analysis of the kinetics of [³H]ʟ-alanine uptake in BasC PLs when varying the external concentration of alanine in exchange with intraliposomal 4 mM ʟ-alanine revealed two components: one in the high (μM) and the other in the low (mM) range of apparent affinity (Fig. 2b). Notably, the addition of Nb74 abolished the low apparent affinity component (Fig. 2b). These results demonstrate unequivocally that the apparent high affinity component corresponds to the extracellular side of the transporter. Moreover, the extracellular $K_m$ for [³H]ʟ-alanine determined with ʟ-alanine concentrations up to 800 μM in the absence of Nb74 showed similar values to the full range kinetics inhibited by 5 μM Nb74 (65 ± 12 and 46 ± 9 μM, respectively) (Fig. 2b and Supplementary Fig. 4c).

**The substrate binding site.** The solved apo BasC structure corresponds to inward facing conformation with the substrate cavity open to the cytoplasmic side, similar to the previously determined apo inward-facing GadC (PDB 4DJI)[22] (Supplementary Fig. 5a). Nevertheless, in contrast to BasC, the GadC structure showed a pH-regulated C-plug domain occupying most of the intracellular vestibule[22], hindering the identification of potentially relevant gating and substrate-interacting residues. To define the substrate-binding site of BasC, we crystallized the BasC-Nb74 complex in the presence of 2-AIB. This amino acid analog is a substrate with low apparent affinity (intracellular $K_m$ 34.5 ± 10 mM; i.e., ~10-fold higher than ʟ-alanine $K_m$) (Supplementary Fig. 6). Extra

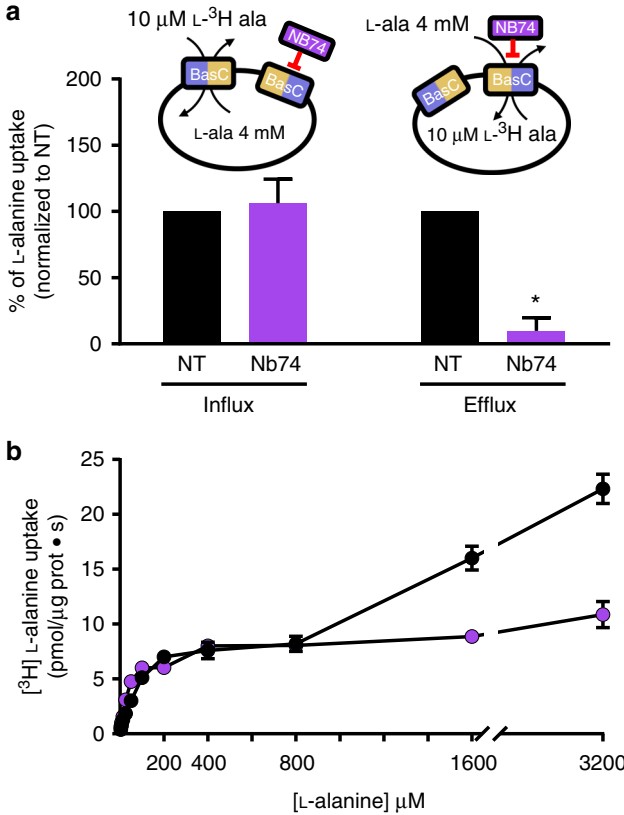

**Fig. 2** Nb74 reveals the sidedness of the substrate interaction. **a** The addition of Nb74 (5 µM) to the medium blocks efflux, but not influx, of 10 µM [³H]L-alanine in exchange with 4 mM L-alanine. Because BasC is randomly inserted in proteoliposomes (PLs) (i.e., right-side-out and inside-out) and Nb74 binds to the cytoplasmic side of BasC, transport of 10 µM [³H]L-alanine occurs from the extracellular side of BasC. Data (mean ± s.e.m.) normalized to non-treated PLs from 5 to 6 independent experiments. *$p < 0.0001$ Student's test vs. NT (non-treated with Nb74). **b** Representative kinetics varying the external concentration of [³H]L-alanine in exchange with 4 mM L-alanine inside PLs. In the absence of Nb74 (black circles), the kinetics is complex without saturation at mM concentrations of extraliposomal L-alanine. Nb74 (magenta circles) converts this kinetics in the single component of high apparent affinity corresponding to the extracellular side of BasC. Data (mean ± s.e.m.) correspond to triplicates. Source data are provided as a Source Data file

electron density was observed in the structure for the bound 2-AIB within the unwound segments of TM1 and TM6 (Polder and electron densities are shown in Figs. 3a, 4b, respectively). This scenario places 2-AIB to the rear of the vestibule, situated approximately in the middle of the plane of the membrane, and with full access to the cytosol (Fig. 1c, d).

2-AIB interacts mainly with backbone atoms from the unwound segments of TM1 and TM6 (Fig. 3a). The α-carboxyl of 2-AIB forms hydrogen bonds with the N-atoms of TM1 residues Ala 20 and Gly 21, and the α-amino group forms hydrogen bonds with carbonyls of residues Val 17 (TM1), and Phe 199, Ala 200 and Asp 202 (TM6). In addition, the hydroxyl group of Tyr 236 (TM7) is at H-bond distance from the carboxylate group of 2-AIB, and the lateral chain of Phe 199 (TM6a) occludes substrate interaction with the D-α-methyl group of 2-AIB (Fig. 3a). To test whether the α-amino and the α-carboxyl moieties of the substrate are essential requirements, transport of two-carbon molecules lacking either the α-amino or the α-carboxyl group were evaluated. The exchange of 10 µM

[³H]L-alanine by 4 mM acetate or ethylamine inside the PLs was not significant, in contrast to L-alanine or 2-AIB (Fig. 3b). Moreover, γ-aminobutyrate was not a substrate (Fig. 3b), thus supporting the notion that the α-amino and the α-carboxyl moieties are a substrate requirement of BasC, as has been shown for human LAT1 and mouse LAT2[26,29,30].

Comparison of the apo and holo structures showed that 2-AIB binding causes a 2 Å displacement of residue Gly 19, located within the unwound segment of TM1 (Fig. 3c). This substrate-induced shift brings the O atom of Gly 19 at H-bond distance from the hydroxyl group of Ser 282 (TM8) (Fig. 3c). In agreement, MD of the holo structure showed that dissociation of the substrate is accompanied by a concomitant relaxation of the TM1 unwound segment that separates Gly 19 and Ser 282, and reduces the distance between Gly 19 and Asp 202 (Supplementary Fig. 7a–e), thereby supporting the notion of a substrate-induced fit. Ser 282 is fully conserved among human LATs[3] (Supplementary Fig. 8), although mutation to alanine in both BasC (S282A) and human Asc-1 (hAsc-1) (S325A) did not abolish 10 µM [³H]L-alanine uptake (Fig. 3d). These results indicate that the interaction of TM1 (Gly 19) and TM8 (Ser 282) induced by substrate binding is not a requisite for transport function. In agreement, MD showed that the Gly 19–Ser 282 H-bond is not stable, due to rotamer oscillation of Ser 282 while 2-AIB is in the binding site (Supplementary Fig. 7e).

**Mutational analysis of the BasC binding site.** Although the recognition of 2-AIB by the cytoplasmic face of the transporter is defined mainly by interactions with backbone atoms of BasC, crystal structures revealed Tyr 236 as a possible substrate interactor (Fig. 3). In the apo structure, the hydroxyl group of Tyr 236 is at H-bond distance from the backbone N of Gly 19 (TM1 unwound segment) and from the backbone O of Ala 200 (TM6a) (Fig. 4a). The displacement of Gly 19 caused by the 2-AIB-induced fit, as seen in the structure of the BasC-2-AIB complex, rearranges the potential H-bonds of the hydroxyl group of Tyr 236 towards the backbone O of Ala 200, and maintains the carboxyl group of the substrate at H-bond distance (Fig. 4b). Of note, Tyr 236, which is fully conserved among human LATs[3], sits in the same position as the sodium ion in the sodium-one (Na1) site in sodium-dependent APC superfamily transporters, where the cation participates in substrate binding (Supplementary Fig. 9 and ref. [3,24]). To question whether the hydroxyl group of Tyr 236 participates in substrate binding in the sodium-independent transporter BasC, we examined the transport function of the Y236F mutant in reconstituted PLs. A small increase of 10 µM [³H]L-alanine uptake was observed (Fig. 4c). Functional characterization of the homologous mutant in the hAsc-1 transporter (Y280F) in HeLa cells showed an almost identical increment in 10 µM [³H]L-alanine uptake (Fig. 4c), thereby suggesting that this residue has a similar role in both BasC and human transporters. Kinetic characterization of the BasC mutant Y236F revealed an increased cytoplasmic apparent affinity for L-alanine (~4-fold vs. wild-type BasC), but unaffected extracellular apparent affinity (Table 1), supporting the notion that Tyr 236 participates in the asymmetry of apparent substrate affinity, a characteristic of both BasC and human LATs[3,5,31]. Thermostability-based binding assays[25,32] of wild-type BasC and the Y236F mutant suggested no significant differences in apparent $K_D$ values for L-alanine (Supplementary Fig. 10a). In agreement, MD indicated that the interaction of Tyr 236 (OH) with 2-AIB (COO) is not stable, and in contrast to the stable 2-AIB (N)—Ala 200 (O) distance, the distance between the O atoms of Tyr 236 and 2-AIB, rapidly increased from 3.2 Å (Fig. 4a) to ≥4 Å (Supplementary Fig. 7f, g). As Tyr 236 would make a small contribution to substrate binding,

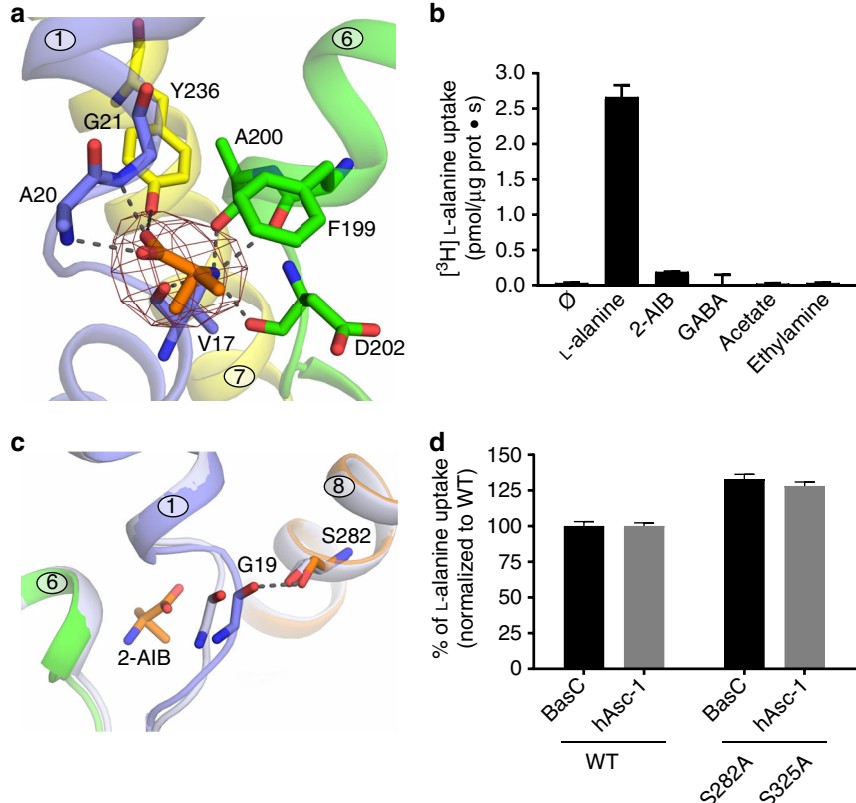

**Fig. 3** Structure of the amino acid binding site and substrate-induced fitting. **a** View of the bound 2-AIB ligand, showing the POLDER electron density map contoured at 3σ, dark red. Distances between atoms of the substrate and BasC residues compatible with H bonds are indicated (dashed lines). **b** 10 μM [³H]ʟ-alanine exchange (pmol μg protein⁻¹ s⁻¹) into BasC PLs containing 4 mM of the indicated compounds. **c** Binding of 2-AIB displaces Gly 19 in the BasC apo structure (light gray) to H bond distance (dashed line) with Ser 282 in the holo structure (rainbow). **d** 10 μM [³H]ʟ-alanine/4 mM ʟ-alanine exchange by BasC reconstituted in proteoliposomes (PLs) (black bars) and transport of 10 μM [³H]ʟ-alanine in HeLa cells by human Asc-1 (hAsc-1) (gray bars). Wild-type (WT) and the indicated homologous mutants of BasC and hAsc-1 were studied. Transport is normalized to the corresponding wild-type values. In **b, d**, data (mean ± s.e.m.) from 3 independent experiments are shown. Source data are provided as a Source Data file

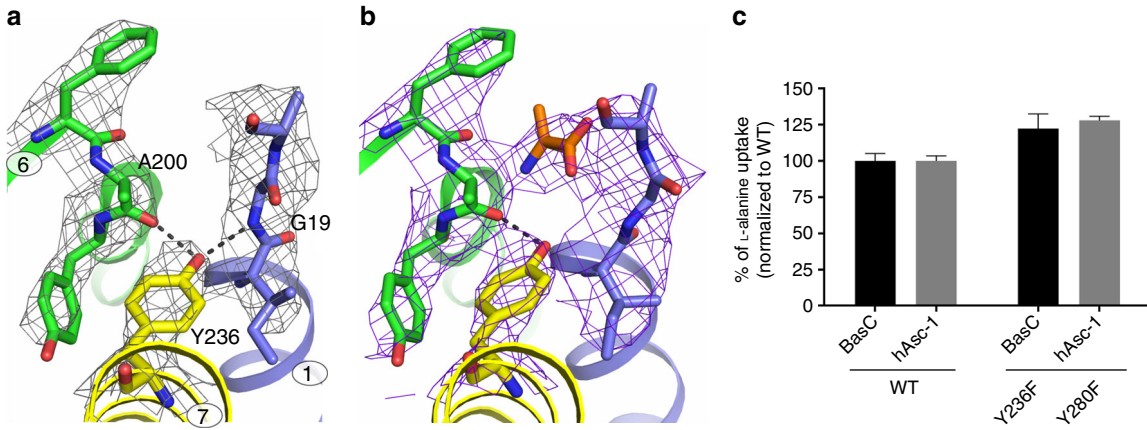

**Fig. 4** Structural and functional interactions of Tyr 236. 2Fo-Fc density maps contoured at 1σ for the BasC binding site residues in apo at 2.9 Å (gray) (**a**) and 2-AIB bound at 3.4 Å (purple) (**b**) structures. **c** 10 μM [³H]ʟ-alanine/4 mM ʟ-alanine exchange by wild-type (WT) and Y236F mutant BasC reconstituted in proteoliposomes (black bars) and transport of 10 μM [³H]ʟ-alanine in HeLa cells by human Asc-1 (hAsc-1) WT and Y280F mutant (gray bars). Data (mean ± s.e.m.) normalized to WT values from 3 independent experiments. Source data are provided as a Source Data file

the increased cytoplasmic apparent affinity of the Y236F mutant is likely to be the result of altered transitions during the transport cycle.

**Determinants of the apparent substrate affinity asymmetry.** As Tyr 236, located in a position equivalent to the Na1 site of

sodium-dependent APC superfamily transporters (Supplementary Fig. 9), is partially responsible for the substrate-interaction asymmetry of BasC, we next focused on the Na2 site. Sodium binding to the Na2 site is associated with increased substrate binding affinity[33,34]. The BasC residue Lys 154 (TM5) is located in an equivalent position to the Na2 site (Fig. 5a and

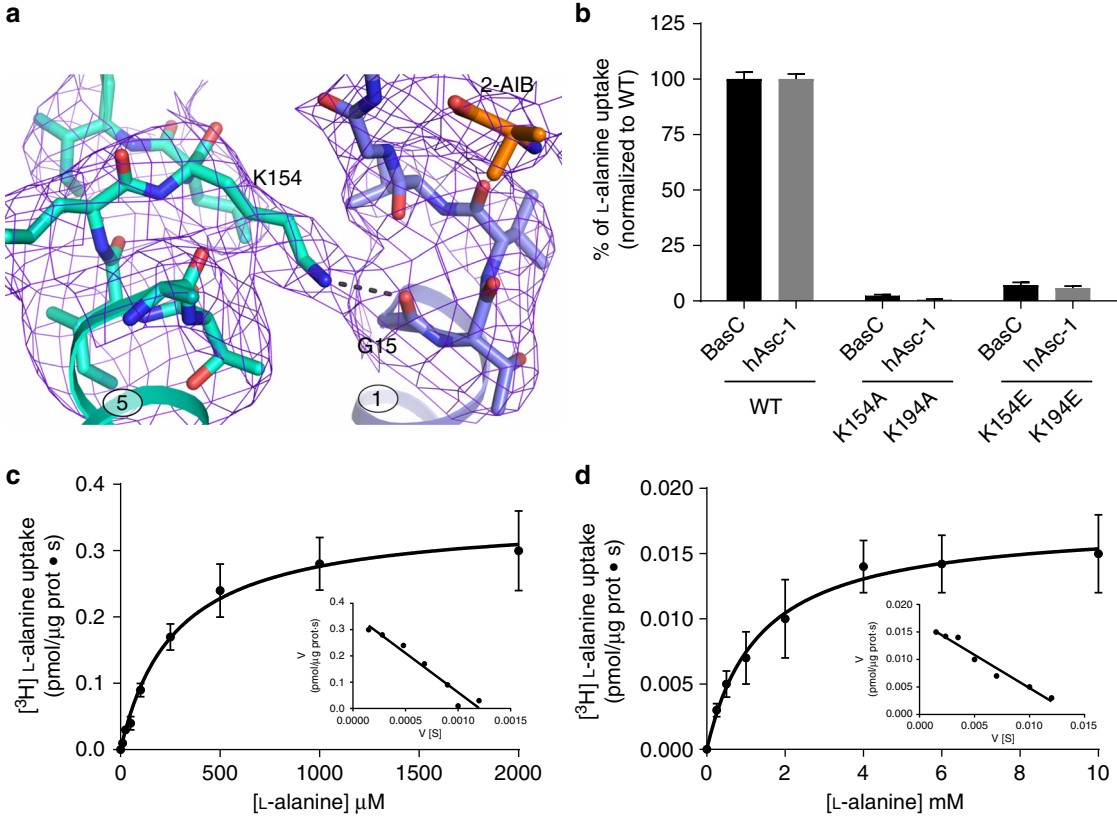

**Fig. 5** Role of Lys 154 in the asymmetry of the substrate interaction. **a** Lys 154 forms an H bond with Gly 15 at the C-terminal end of TM 1a in the BasC holo structure. 2Fo-Fc electron density map (blue) is shown. **b** 10 μM [³H]L-alanine/4 mM L-alanine exchange by BasC reconstituted in proteoliposomes (PLs) (black bars) and transport of 10 μM [³H]L-alanine in HeLa cells by human Asc-1 (hAsc-1) (gray bars). Wild-type (WT) and the indicated homologous mutants of BasC and human Asc-1 (hAsc-1) were studied. In **b**, data (mean ± s.e.m.) normalized to WT values from 3 independent experiments are shown. Representative kinetic experiments of either extracellular (**c**) or cytoplasmic (**d**) sides of K154A BasC mutant reconstituted in PLs. In both cases, kinetics was examined in the presence of 5 μM Nb74 to monitor only right-side-out-oriented transporters. In **c**, **d**, data (mean ± s.e.m.) correspond to quadruplicates. Eadie-Hofstee transformation of the kinetics presented in **c** and **d** is shown in the insets. Source data are provided as a Source Data file

Supplementary Fig. 9). Lys 154 is fully conserved among human LATs and BasC[3]. The ε-amino group of Lys 154 points towards the carbonyl group of Gly 15 and the electron density map unambiguously showed the interaction of Lys 154 with Gly 15 at the C-terminal end of TM1a in the holo (Fig. 5a) structure of BasC.

Mutation K154A in BasC and the homologous mutation K194A in hAsc-1 almost completely abolished 10 μM [³H] L-alanine uptake (Fig. 5b). Kinetic analysis of L-alanine uptake in the BasC K154A mutant revealed a dramatic reduction (~10-fold vs. wild-type) of both the extracellular apparent affinity and $V_{max}$ (Table 1), with no impact on the intracellular apparent affinity for L-alanine (Fig. 5c, d and Table 1). Similarly, mutation K194A in hAsc-1 dramatically reduced (~10-fold vs. wild-type) both the external apparent affinity ($K_m$ values of 67.5 ± 4.2 and 825 ± 53 μM in wild-type and mutant, respectively) and the $V_{max}$ (1703 ± 425 and 152 ± 16 pmols μg protein⁻¹ min⁻¹ in wild-type and mutant, respectively) of L-alanine uptake. Interestingly, a lysine to glutamate change (K154E in BasC, K194E in hAsc-1), a mutation that in human y⁺LAT1 (K191E) causes lysinuric protein intolerance[35], also dramatically reduced 10 μM [³H]L-alanine uptake in both BasC and hAsc-1 (Fig. 5b). Overall, these data strongly suggest that this lysine is key for BasC and hAsc-1 function and also for the asymmetric interaction of the substrate at both sides of the transporter, by supporting high apparent affinity of BasC for L-alanine at the extracellular side. Thus, the K154A mutation turns BasC into a more symmetric

transporter with apparent $K_m$ in the mM or near mM range at both sides of the membrane. Additionally, the highly reduced $V_{max}$ observed in BasC K154A and in the homologous mutant K194A of hAsc-1 suggests that this lysine also participates in key steps of the transport cycle.

MD gave an unexpected clue as to the role of Lys 154. Unbiased MD of the structure of the BasC-2-AIB complex showed substrate dissociation from the substrate-binding site to simultaneously bind Lys 154 (O atom of 2-AIB with the ε-amino group of Lys 154) and Thr 16 (N atom of 2-AIB with the backbone O atom of Thr 16) (Fig. 6 and Supplementary Fig. 7j). These results suggest that 2-AIB-Lys 154 is the first transient interaction of the substrate as it moves to the cytosol. Supporting this hypothesis, thermostability-based binding assays indicated a small but significant reduction in the K154A L-alanine binding affinity compared with that of the wild-type (Supplementary Fig. 10a), thereby reinforcing the role of this residue in substrate interaction.

## Discussion

Here we present the structures of a LAT subfamily member (BasC) in the apo- and holo-form. BasC crystallized in the presence of a nanobody (Nb74) that recognizes the intracellular region of the transporter, demonstrating the sidedness of the substrate apparent affinities of the transporter. The BasC apo and 2-AIB-bound structures in non-occluded inward-facing state

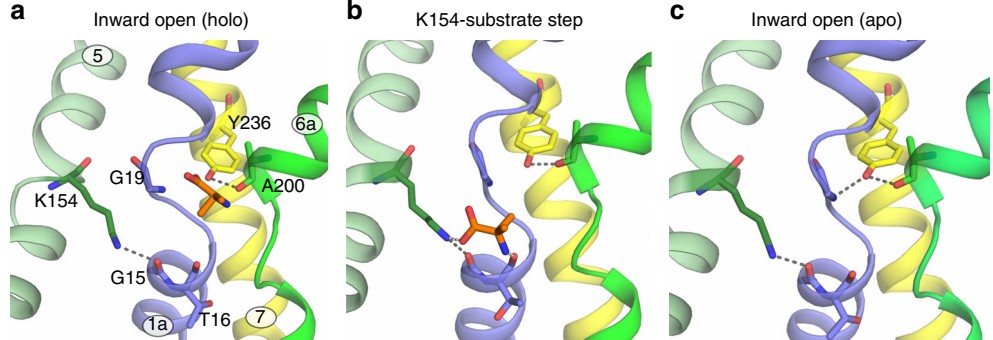

**Fig. 6** Model for BasC substrate release to the cytoplasm. **a** 2-AIB (orange) bound BasC structure (PDB ID: 6F2W). Interactions of Lys 154 in TM5 with Gly 15 in TM1a and Tyr 236 in TM7 with Ala 200 in TM6 are shown. **b** Molecular dynamics analysis shows interaction of 2-AIB with the amino group of Lys 154 and the (O) backbone atom of Thr 16 in TM1a in the way of the substrate to the cytosol (snapshot extracted at 435 ns from the blue trajectory shown in Supplementary Fig. 7). **c** In the BasC apo structure (PDB ID: 6F2G), Gly 19 in the unwound region of TM1 approaches Tyr 236 at hydrogen bond distance. Lys 154-Gly 15 interaction is maintained all along the MD analysis and in both apo and holo structures

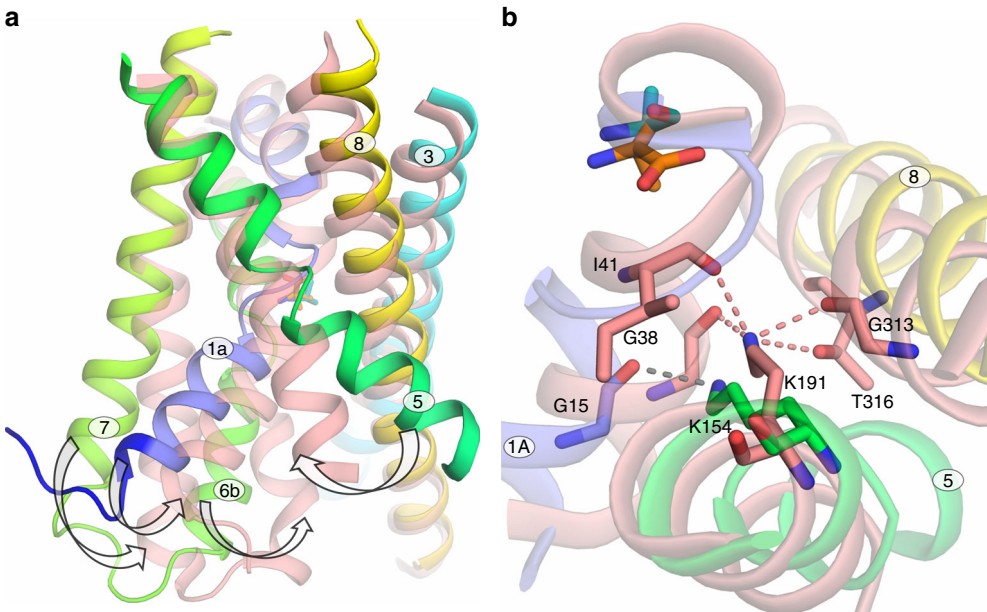

**Fig. 7** Substrate occlusion in inward-facing SLC7 transporters. **a** Comparison of the structures of BasC (rainbow) bound to 2-AIB (orange) (PDB ID: 6F2W) in non-occluded state and GkApcT (salmon) bound to L-alanine (cyan) (PDB ID: 5OQT) in occluded state. Tilting of TM1a and TM6b, together with the accompanying TM7, closes the substrate vestibule to occlude the substrate at the cytoplasmic side (arrows). Concomitantly, TM1a interacts and attracts TM5 in the occlusion state (arrow). Only TM1, TM3, TM5, TM6, TM7, and TM8 are depicted for clarity. **b** Detail of the interactions of K154 in BasC (gray dashed line) and K191 in GkApcT (salmon dashed lines) in the structures shown in **a**

offer the first clues on the access (and release) of the substrate from the cytosol and the substrate-induced fit in LATs. Finally, two fully conserved residues in LATs, namely Tyr 236 (TM7) and Lys 154 (TM5) in BasC, are responsible for the asymmetry of the apparent affinity of the substrate, a key feature in the physiological role of LATs.

The non-occluded inward-facing conformation structure of BasC in complex with 2-AIB fills a gap in the knowledge of the transport cycle of the APC transporter family[17–23,36]. Additionally, structural superimposition of BasC with other APC superfamily transporters in the same conformation, shows that, although similar, differences can be observed, particularly in the tilting of TMs 1a, 5, and 7 (Supplementary Fig. 5). Comparison with the occluded inward-facing structure of the bacterial CAT homolog GkApcT (with substrate bound) (PDB ID 5OQT) sheds new light on the access of the substrate from the cytosol. Thus, tilting of TM1a and TM6b, which are more open in BasC

(non-occluded), and the concomitant dissociation of TM1a from TM5, facilitate access of the substrate from the cytosol to the substrate-binding site and vice versa (Fig. 7).

Direct interactions of the α-amino and the α-carboxyl moieties of the substrate with the unwound segments of TM1 and TM6 explain the binding of 2-AIB with BasC (Figs. 3a, 4b), a similar design to that previously reported for structures of amino acid transporters of the APC superfamily in inward-facing conformations[23,37,38]. Nevertheless, small differences could be found when compared with other APC-fold transporters. Indeed, in GkApcT, a water molecule connects the substrate with backbone atoms of residues in the unwound segment of TM1 and the hydroxyl group of Tyr 268 (TM7). This suggests a slightly different substrate binding recognition between CAT and LAT subfamilies or a modification of the binding site upon occlusion[23]. Additionally, in contrast to AdiC, the α-amino and α-carboxyl moieties of the substrate are a requirement for BasC

transport activity[36] (Fig. 3d). This feature parallels transport requirements in mammalian LAT1 and LAT2, where the α-amino and the α-carboxyl (or a modified carboxyl) groups are required for transport[26,29,30].

TM1–TM8 coordination is key for the transport function in APC superfamily transporters. Indeed, in sodium-dependent members, one $Na^+$ ion (named Na2) binds the unwound region of TM1 in bundle domains and TM8 in hash domains (Supplementary Fig. 9), resulting in increased substrate binding affinity and conformational transitions[34,39,40]. Similarly, in the sodium-independent APC superfamily transporter CaiT, the Arg 262 side chain in TM5 has been proposed to substitute the function of Na2 by mediating TM1–TM8 coordination during the translocation cycle[41]. In contrast, in the sodium-independent arginine/agmatine exchanger AdiC, which lacks a positively charged residue in the putative Na2 site, the substrate guanidinium group interacts with an aromatic side chain (Trp 293) in TM8[17,36] connecting TM1–TM8. In this regard, mutations of residues in TM8 involved in TM1–TM8 coordination in APC superfamily transporters result in a strong decrease in both substrate affinity and transporter activity[33,34,41]. In a similar way, mutation of BasC Lys 154 in TM5, a residue lined with the Na2 site (Supplementary Fig. 9), in addition to decreasing by ~10-fold the external apparent affinity of the substrate L-alanine, also decreases dramatically the maximal transport activity (~1/10 of wild-type activity). Superimposition of BasC and GkApcT structures (Fig. 7a) reveals that TM1, TM5, and TM8 movement during inward-open to inward-occluded transition sets Lys 154 (BasC numbering) at hydrogen bond distance with TM8 residues (Fig. 7b)[23]. Indeed, the GkApcT electron density map shows an interaction between Lys 191 (Lys 154 in BasC) and backbone atoms of Gly 313 (Ser 282 in BasC), reinforcing the idea that the hydroxyl group of Ser 282 is not essential for transporter activity (Fig. 3d). Overall, these results strongly suggest that TM1–TM5 (Gly 15–Lys 154) (Figs. 5a, 6) is important for the inward-open conformation, while inward-open to inward-occluded transition results in TM1–TM5–TM8 coordination, similarly to other APC superfamily transporters.

A physiologically relevant characteristic of LATs is the asymmetry in the apparent substrate affinity at both the intracellular and extracellular sides of the transporter. This asymmetry allows LAT transporters to control intracellular amino acid pools (mM concentrations) by exchange with external amino acids (μM concentration range)[4,5]. In a similar manner to human LAT1, LAT2 and b[0,+]AT[5,31], BasC also shows this asymmetry, as demonstrated previously by vectorial inactivation of the transport activity from the extracellular side of BasC[3], and here by blocking the transport activity with Nb74 upon binding to the cytoplasmic side of the transporter. In this context, the interaction of Nb74 with BasC via residues from TM6b in bundle domains and from TM8 and TM9 in hash domains explains the inhibitory effect of Nb74 on BasC transport activity. Indeed, based on the rocking bundle translocation mechanism proposed for APC superfamily transporters, tilting of the bundle domain over the hash domain rationalizes the major conformational changes in inward-outward facing transitions[42]. In this regard, by specifically inhibiting the inside-out BasC molecules in PLs, Nb74 emerges as an excellent tool with which to study the asymmetric substrate interaction of BasC (Fig. 2b). Under these conditions, the functionally remaining right-side-out-oriented transporters clearly revealed the high apparent affinity of the extracellular side of BasC for substrates (Fig. 2b and Supplementary Fig. 5c).

Two fully conserved residues in LATs, Tyr 236 (TM7), and Lys 154 (TM5) in BasC, are responsible for this asymmetry, as mutations Y236F and K154A alter the internal and external $K_m$ for L-alanine to ~500 μM (Table 1). Binding assays suggest that K154A and Y236F alter the apparent affinity for L-alanine by

affecting mainly key steps in the transport cycle that have an impact on the extracellular or cytoplasmic $K_m$ values, as previously proposed for the plant Major Facilitator Superfamily nitrate transporter NRT1.1[43]. Nevertheless, the molecular bases for these changes are unknown; however, both structural and MD analysis provided several clues. On the one hand, the robust H-bond between BasC residues Tyr 236 and Ala 200 (TM6a) (Fig. 4a and Supplementary Fig. 7h) and the substrate-induced weakening of the Tyr 236–Gly 19 H-bond (Figs. 4b, 6 and Supplementary Fig. 7i) might be responsible for maintaining the differential tension on the unwound segments of TM1 and TM6 in order to induce transitions to the next steps in the transport cycle. In this regard, Tyr 239, the corresponding residue in AdiC of BasC Tyr 236, (Supplementary Fig. 8), could play a role in the moderate apparent affinity asymmetry reported for both arginine and agmatine (i.e., two-fold higher intracellular apparent Km value compared to the extracellular one)[44]. In fact, the reported asymmetry for agmatine, which lacks the α-carboxyl group, reinforces the idea that, similarly to Tyr 236 in BasC, interaction of Tyr 239 hydroxyl group with substrate α-carboxyl is not essential for the establishment of substrate apparent affinity asymmetry. Nevertheless, determination of substrate $K_D$ at both sides of the transporter will be necessary to define the thermodynamic and kinetic contributions to the decreased intracellular apparent $K_m$ of Y236F mutant.

On the other hand, MD trajectories of the BasC-2-AIB complex revealed binding of the substrate to Lys 154 (Supplementary Fig. 7j), thereby suggesting that this residue facilitates the release of the substrate to the cytosol (Fig. 6). In this context, the K154A mutation significantly reduces L-alanine binding. Nevertheless, the dramatic impact of the K154A mutation on extracellular substrate apparent binding affinity and $V_{max}$ suggests that Lys 154 also facilitates the outward-to-inward transition, thereby increasing the apparent affinity for substrates at the extracellular face. Similarly, mutation of Lys 191 in GkApcT and Lys 158 in ApcT, which are located in the same position as Lys 154 in BasC (Supplementary Fig. 8), results in a nearly complete reduction of L-alanine uptake, suggesting a similar role for these residues in transporter function although their effects on substrate apparent affinity have not been addressed[21,23].

In summary, we present the crystal structures of a LAT transporter. These structures should enable the building of robust models of human LATs, which in turn can facilitate the generation of specific inhibitors to target transporters of therapeutic interest[45–47]. Moreover, the functional characteristics of BasC, in line with those of human LATs, make this transporter a useful model to decipher the molecular mechanisms of LATs and to reveal the molecular defects underlying pathologic mutations in human LATs. In this regard, the K154 mutation reported here is a good example of a residue that is mutated in human disease (lysinuric protein intolerance) and whose underlying molecular defect can be uncovered by the functional and structural study of BasC.

## Methods

**BasC expression in *E. coli* and membrane preparation.** BasC was overexpressed as a C-terminal fusion with GFP in *E. coli* BL21 Star (DE3) cells grown in LB media. Single point mutations (Supplementary Table 2) were introduced using the QuikChange site-directed mutagenesis kit (Stratagene, San Diego, CA). All mutations were verified by sequencing. Expression was induced with 0.1 mM isopropyl-ß-D-thiogalactopyranoside at 37 °C for 22 h. Cells were harvested at 5000 × g for 15 min at 4 °C and stored at −80 °C until use. Cell pellets were thawed and resuspended in 20 mM Tris-Base, 150 mM NaCl, pH 7.4. Cells were pelleted again at 5000 × g for 15 min at 4 °C, resuspended in lysis buffer (20 mM Tris-Base, 350 mM NaCl, pH 7.4, 1 mM pefabloc and complete mini protease inhibitor cocktail) (Roche, Basel Switzerland) and then broken using a Cell Disruptor (four cycles at 20,000 psi; Constant Systems Ltd., Daventry, UK). Cell debris was removed by centrifugation (15,000 × g for 1 h at 4 °C), and the supernatant was

subjected to ultracentrifugation (200,000 × *g* for 2 h at 4 °C). The membrane pellet was resuspended in 20 mM Tris-Base, 150 mM NaCl, pH 7.4 and 10% glycerol at a protein concentration of 8–12 mg ml⁻¹, and then frozen in liquid nitrogen and stored at −80 °C until use.

**BasC purification for crystallography.** All subsequent steps were carried out at 4 °C. Membranes (3 mg ml⁻¹ protein concentration) were solubilized using 2% (w/v) n-decyl-β-D-maltopyranoside (DM; Affymetrix, Santa Clara, CA) for 1 h in purification buffer (20 mM Tris-Base, 150 mM NaCl, pH 7.4 and 10% glycerol). Following ultracentrifugation (200,000 × *g* for 2 h), the soluble fraction was incubated for 3 h with Ni²⁺-NTA Superflow beads (Qiagen, Hilden, Germany) equilibrated in washing buffer (20 mM Tris-Base, 150 mM NaCl, pH 7.4, 0.17% DM, 10% glycerol and 20 mM imidazole). Protein-bound beads were washed three times with 20 column volumes of washing buffer before on-column cleavage with HRV-3C protease (IRB Barcelona Protein Expression Core Facility, Barcelona, Spain) for 16 h. Column flow through containing cleaved BasC was concentrated by centrifugation in an Amicon Ultra-15 filter unit (100,000 kDa molecular weight cut-off; Millipore, Temecula, CA) at 3220 × *g* until reaching 6 mg ml⁻¹ protein and then incubated overnight with nanobody 74 (Nb74) at a molar ratio of 1:1.2 (BasC:Nb74). The complex was subjected to size exclusion chromatography (SEC) on a Superdex 200 10/300 GL column (GE Healthcare, Chicago, IL) equilibrated with 20 mM Tris-Base, 150 mM NaCl, pH 7.4 and 0.17% DM. Next, 200-μl fractions were collected and used for crystallization. For the purification of 2-aminoisobutyric acid (2-AIB)-bound protein, all buffers contained 100 mM 2-AIB.

To prepare selenomethionine (SeMet)-labeled BasC protein, *E. coli* cells from 100 ml of an overnight LB culture were pelleted, washed, resuspended in 1 ml of SelenoMet medium (Molecular Dimensions Ltd., Newmarket, UK), and then inoculated into 1 l of pre-warmed (37 °C) SelenoMet medium containing 40 μg ml⁻¹ of L-SeMet (Sigma-Aldrich, Madrid, Spain). SeMet-labeled BasC protein was expressed and purified as for the unlabeled protein.

**Generation and characterization of nanobodies against BasC.** Nanobodies (Nbs) were prepared against the wild-type BasC protein as described previously[48]. In brief, a llama (*Lama glama*) received six weekly injections of 100 μg of purified BasC reconstituted in *E. coli* polar lipid proteoliposomes (PLs) at a protein to lipid ratio of 1:50. The Nb-encoding ORFs were amplified from total lymphocyte RNA and subcloned into the phage display/expression vector pMESy4. After one round of panning, clear enrichment was seen for the BasC protein. Subsequently, 88 individual colonies were randomly picked, and the Nbs were produced as soluble His- and CaptureSelect C-tagged proteins (MW 12–15 kDa) in the periplasm of *E. coli*. Testing for specific BasC protein binding resulted in 29 families. All selections and screenings were done in the absence and presence of L-alanine. Inducible periplasmic expression of Nbs in *E. coli* WK6 strain produced milligram amounts of >95% pure Nb using immobilized Co²⁺ ion affinity chromatography (Talon resin; Takara Bio Inc, Kusatsu, Japan) from the periplasmic extract of a 1-l culture. Purified Nbs (2–10 mg ml⁻¹) in 20 mM Tris-Base, NaCl 150 mM, pH 7.4 were frozen in liquid nitrogen and stored at −80 °C before use.

Surface plasmon resonance (Biacore T-100, GE Healthcare, Chicago, IL) was used to screen 29 Nbs (one from each family) for binding with BasC, purified as for crystallography. The Nbs (3–20 μg ml⁻¹) were immobilized to reach around 200 resonance units (RU) (injections of 20 s at 5 μl min⁻¹) on a CM5 Sensor Chip previously coated with an anti_His antibody (His capture kit; GE Healthcare, Chicago, IL) and regenerated with 10 mM HCl in running buffer (phosphate buffered saline, 0.17% DM). BasC was assayed at 500–2.5 nM in running buffer. The six Nbs with the highest Kd values (up to 30 nM) entered the crystallization screenings. The BasC-Nb74 complex rendered the best results initially and consequently crystals were optimized.

**BasC crystallization and data collection strategy.** Purified BasC:Nb74, BasC:Nb74-2-AIB and SeMet-labeled BasC:Nb74 complexes were concentrated to 1.2 mg ml⁻¹ and filtered by centrifugation on a Spin-X centrifuge tube filter (Corning Inc, Salt Lake City, UT) at 3220 × *g*. For high lipid detergent (HiLiDe) crystallization, BasC was concentrated to 10 mg ml⁻¹ and incubated overnight at 4 °C in a mixture of n-dodecyl-β-D-maltopyranoside (DDM; Affymetrix, Santa Clara, CA) and palmitoyl-2-oleoyl-sn-glycerol-3-phosphocholine (POPC) (protein:lipid: detergent; w-w:w: 1:0.04:0.17)[49]. For re-lipidation, 100 μl of 10 mg ml⁻¹ BasC protein sample was added to a glass tube pretreated with a thin film of POPC. The film was prepared by dispensing POPC dissolved in CHCl₃, followed by evaporation of the CHCl₃ using nitrogen gas (N₂) at room temperature. Subsequently, the lipidated protein samples were supplemented with 1.7 mg of (extra) DDM and the tubes were stirred at 50 rpm using microstirring bars (5 × 2 mm) for 18 h at 4 °C. The insoluble material was subsequently removed by ultracentrifugation at 190,000 × *g* for 10 min and the transparent supernatants were used for crystallization experiments. Crystals were grown by sitting-drop vapor diffusion at 4 °C by mixing equal volumes of protein and reservoir solution containing 27–29% PEG 400 and 0.1 M ammonium acetate pH 8 (0.1 M glycine pH 9 for HiLiDe crystals). Crystals typically appeared after 4 days, reaching a maximum size after 10–14 days. Crystals were cryoprotected by soaking in 33% PEG 400 and then flash-cooled in liquid nitrogen and diffracted at the ALBA synchrotron light source (Cerdanyola

del Vallès, Spain) and at the European Synchrotron Radiation Facility (Grenoble, France). Special care was taken in data collection to overcome the inherent crystal anisotropy. To this end, crystals were aligned to minimize this effect by using tilted loops and a mini-kappa (MK3) with automatic recentering. The best crystals with and without 2-AIB diffracted up to 3.4 and 2.9 Å, respectively. Complete datasets of SeMet-labeled BasC were obtained up to 4.2 Å.

**Structure determination.** Data were processed with Xia2[50] using XDS[51], Aimless and Pointless[52] from the CCP4i suite of programs[53]. The Diffraction Anisotropy Server from UCLA was used for anisotropic scaling[54]. Phases were obtained by molecular replacement using PHASER[55] and the structures of ApcT (PDB ID: 3GIA) and GadC (4DJI) and a nanobody (5H8D) as templates. Additionally, anomalous data from a modified SeMet protein crystal were collected and methionines were traced in the model. The structure of the Nb74-BasC-2-AIB complex was solved by molecular replacement based on the BasC-Nb74 apo structure. Model building into the electron density map was performed in COOT[56], with structure refinement carried out in autoBUSTER[57] and REFMAC[58]. Polder and Omit maps were calculated with Phenix[59]. Images were prepared using Open-Source PyMol (The PyMOL Molecular Graphics System, Version 2.0 Schrödinger, LLC.).

**BasC purification for amino acid transport assays.** All subsequent steps were performed at 4 °C. Membranes (3 mg ml⁻¹ protein concentration) from *E. coli* expressing BasC-GFP were solubilized using 1% (w/v) DDM for 1 h in purification buffer (20 mM Tris-Base, 150 mM NaCl, pH 7.4 and 10% glycerol). After ultracentrifugation (200,000 × *g* for 2 h), the soluble fraction was incubated for 3 h at 4 °C with Ni²⁺-NTA Superflow beads equilibrated with washing buffer (20 mM Tris-Base, 150 mM NaCl, pH 7.4, 0.05% DDM, 10% glycerol and 20 mM imidazole). Protein-bound beads were washed three times with 10 column volumes of washing buffer before elution with washing buffer supplemented with 350 mM imidazole. The purified protein was concentrated by centrifugation in an Amicon Ultra (100,000 kDa molecular weight cut-off; Millipore) at 3220 × *g* until the desired protein concentration was reached.

**Reconstitution into proteoliposomes.** BasC-GFP protein was reconstituted in *E. coli* polar lipids (Sigma-Aldrich, Madrid, Spain), as previously described[3]. Lipids were dried under N₂ and suspended in reconstitution buffer (20 mM Tris-Base, 150 mM NaCl, pH 7.4). The suspension was sonicated to clarity and purified BasC protein was added to reach the desired protein to lipid ratio of 1:100 (w:w). To destabilize the liposomes, 1.25% ß-D-octylglucoside was added and the mixture was incubated on ice with occasional agitation for 5 min. DDM and ß-D-octylglucoside were removed by dialysis for 40 h at 4 °C against 100 volumes of dialysis buffer. Proteoliposome (PL) suspensions were frozen in liquid N₂ and stored at −80 °C until use.

**Amino acid transport assays in proteoliposomes.** For uptake experiments, the desired intraliposomal amino acid concentration was added to the PL suspension, which was then subjected to three freeze/thaw cycles. The extraliposomal amino acid content was then removed by ultracentrifugation (100,000 × *g* for 1 h at 4 °C) and PLs were resuspended to one-third of the initial volume with reconstitution buffer. Amino acid uptake assays were initiated after mixing 10 μl of cold PLs with 180 μl of transport buffer (20 mM Tris-Base, 150 mM NaCl, pH 7.4 plus 0.5–1 μCi/ 180 μl of radiolabeled L-amino acid (Perkin Elmer, Waltham, MA)) supplemented with the unlabeled amino acid to the desired concentration. This mixture was then incubated at room temperature for the indicated periods. Efflux measurements were performed by filling the liposomes with reconstitution buffer plus 10 μM L-alanine and 1 μCi/5 μl of [³H]L-alanine by means of three freeze/thaw cycles. The release of [³H]L-alanine was measured after adding 180 μl of transport buffer with or without 4 mM cold amino acid. Transport experiments were stopped by the addition of 2 ml of ice-cold stop buffer (reconstitution buffer containing or not 5 mM L-alanine for influx or efflux measurements, respectively) and filtration through 0.45-μm pore-size membrane filters (Sartorius Stedim Biotech, Cedex, France). Filters were then washed twice with 2 ml of stop buffer and dried, and the trapped radioactivity was counted. Transport measured in PLs containing no amino acid was subtracted from each data point to calculate the net exchange. Transport values are expressed in pmol of L-alanine per μg of protein and for the indicated time. BasC protein in PLs was determined by Coomassie blue staining in SDS-PAGE gels compared with known amounts of BasC in DDM micelles, determined by BCA assay (Pierce, Rockford, IL) and loaded in the same gel.

Saturation kinetics were analyzed by nonlinear regression analysis, and the kinetic parameters derived from this method were confirmed by linear regression analysis of the derived Eadie-Hofstee plots. Data are expressed as the mean ± s.e.m. of three experiments carried out on different days and on different batches of protein and PLs.

Due to the clear difference in the $K_m$ values at both sides of BasC, most of the saturation kinetics in this work was designed to determine the extraliposomal $K_m$ using a range of [³H]L-alanine up to 250 μM in exchange with 4 mM L-alanine inside the PLs. To determine the intraliposomal $K_m$ values, the exchange of 10 μM [³H]L-alanine with a range of L-alanine concentration up to 10 mM was used.

To determine external and internal $K_m$ values for the K154A mutant, 5 μM Nb74 was added to the K154A PL suspension to block inside-out inserted BasC in the PLs. This procedure allowed us to define the cytosolic kinetic parameters using a larger range of [$^3$H]L-alanine concentrations (up to 2000 mM).

**Substrate binding assays**. GFP-fused BasC proteins purified in DM were incubated for 10 min at 4–60 °C (Y236F and K154A mutants) or 4–70 °C (wild-type). In total, 200 μl of 2 mg ml$^{-1}$ protein was used for each temperature point[60]. Protein was then centrifuged at 186,000 × g for 30 min. Next, 100 μl of the supernatant was loaded onto an SEC column (Superdex 200 5/150 GL) pre-equilibrated with 20 mM Tris-Base, 150 mM NaCl, pH 7.4 and 0.17% DM and run at a flow rate of 0.25 ml min$^{-1}$. Fractions (200 μl) were collected and measured for GFP fluorescence[61]. The peak heights were normalized to those of samples incubated at 4 °C and were fit to the Hill equation using the GraphPad Prism 4 program (GraphPad Software Inc., San Diego, CA). Melting temperatures ($T_m$) were determined by fitting the curves to a sigmoidal dose-response equation, as previously described[61].

To determine the effect of L-alanine on the stability of GFP-fused wild-type BasC, and Y236F and K154A mutants purified in DM, 200 μl of 2 mg ml$^{-1}$ BasC protein was incubated with increasing concentrations of L-alanine (0–200 mM) for 10 min at 4 °C. BasC-GFP proteins were then incubated at the calculated $T_m$ for 10 min and centrifuged at 186,000 × g for 30 min. Next, 100 μl of the supernatant was loaded onto an SEC column (Superdex 200 5/150 GL) pre-equilibrated with 20 mM Tris-Base, 150 mM NaCl, pH 7.4 and 0.17% DM and run at a flow rate of 0.25 ml min$^{-1}$. The peak heights were normalized to those of samples incubated at 4 °C and were plotted vs. L-alanine concentration. Binding affinities were determined by fitting the curves to a sigmoidal dose-response equation.

**Mutagenesis and transfection of wild-type hAsc-1 and mutants**. HeLa cells were maintained at 37 °C in a humidified 5% CO$_2$ environment in DMEM supplemented with 10% fetal bovine serum, 50 units ml$^{-1}$ penicillin, 50 μg ml$^{-1}$ streptomycin and 2 mM L-glutamine. The cells were transiently transfected with the N-terminal c-myc tagged hAsc-1-pRK5 plasmid (a kind gift from Prof. Herman Wolosker; Technion-Israel Institute of Technology) using Lipofectamine 2000 (Invitrogen, Carlsbad, CA). Single point mutations were introduced using the QuikChange mutagenesis kit. All mutations were verified by sequencing. Amino acid transport assays were carried out 24 h after transfection.

**hAsc-1 amino acid transport assays**. Amino acid uptake measurements were performed on hAsc1-transfected HeLa cells. Uptake rates were measured as previously described[45]. Briefly, replicate cultures were incubated with 10 μM cold L-alanine and 1 μCi ml$^{-1}$ [$^3$H]L-alanine at room temperature for 1 min in a sodium-free (137 mM choline chloride) transport buffer that also contained 5 mM KCl, 2 mM CaCl$_2$, 1 mM MgSO$_4$ and 10 mM HEPES (pH 7.4). Uptake was terminated by washing with an excess volume of chilled transport buffer. Saturation kinetics was analyzed by nonlinear regression, and the kinetic parameters derived from this method were confirmed by linear regression analysis of the derived Eadie-Hofstee plots. Data are expressed as the mean ± s.e.m. of three experiments performed on different days and on different batches of cells.

**BasC apo and holo structures molecular dynamics simulations**. The x-ray structure of BasC bound to 2-AIB (holo) was prepared for molecular dynamics simulations with the Protein Preparation Wizard (PrepWizard) tool implemented in Schrödinger[62]. In this regard, first the nanobody, atomic coordinates of water molecules and other cofactors (Zn) were removed. Missing hydrogen atoms were then added by the utility *applyhtreat* in the PrepWizard tool. PROPKA 3.0 was used to calculate the protonation state of titratable residues at pH 7.4 and, on the basis of the predicted p$K_a$ values, the hydrogen bonding network was optimized. The resulting structure was subjected to a restrained minimization step with the OPLSAA force field (FF), keeping heavy atoms in place and optimizing only the positions of the hydrogen atoms.

To model the membrane in the system, BasC coordinates were pre-oriented with respect to the membrane (parallel to the z-axis) by alignment with AdiC (PDB 3OB6) in the OPM database (http://opm.phar.umich.edu)[63]. The protein was then embedded in a POPC lipid bilayer using the CHARMM-GUI Membrane Builder[64–67] by the replacement method. Next, 150 lipid molecules were placed in the lipid bilayer (i.e., 75 lipids in each leaflet) with its center at z = 0. The system was then solvated using a TIP3PM water layer of 20 Å thickness above and below the lipid bilayer. KCl ions corresponding to 0.15 M (40 negative and 31 positive) were also added to the system using Monte Carlo sampling.

In the case of ligand 2-AIB, the automated ligand FF generation procedure (CGenFF) available in CHARMM-GUI was used to generate the FF parameters[68]. Finally, with the CHARMM-GUI Membrane Builder, we also generated the necessary scripts to perform minimization, equilibration and production runs in AMBER, using the CHARMM36 force field (C36 FF), as explained below.

We ran three replicas with different sets of randomly generated initial velocities, using the C36 FF for lipids and the CHARMM TIP3P water model, at constant temperature (300 K) and pressure (1 bar), under Periodic Boundary Conditions, and with Particle Mesh Ewald electrostatics. The simulation time step was set to 2 fs in conjunction with the SHAKE algorithm to constrain the covalent bonds involving hydrogen atoms. After standard Membrane Builder minimization (2.5 ps) and equilibration (375 ps in 6 steps), production simulation was run (500 ns for one trajectory, and 150 ns for another two trajectories).

**Statistical analysis**. Statistical analysis was performed with the Student's t-test using the GraphPad Prism 6 software. Grubb's test ($\alpha = 0.1$) was used to eliminate outliers using the GraphPad Prims 6 software. All the experiments were repeated three or more times.

**Reporting summary**. Further information on research design is available in the Nature Research Reporting Summary linked to this article.

## Data availability

Atomic coordinates for the crystal structures have been deposited in the Protein Data Bank under accession numbers 6F2G (WT-Nb74 complex) and 6F2W (WT-Nb74 2-AIB co-crystal complex). The source data underlying Figs. 2a, b and 3b, d, 4c, 5b–d and Supplementary Figs. 4b, c, 6, and 10a, b are provided as a Source Data file and deposited in Mendeley [https://doi.org/10.17632/7bjsxxzt27.1]. Other data are available from the corresponding authors upon reasonable request.

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

## Acknowledgements

E.E.M. was supported by a Sara Borrell Contract and J.F. and P.B. by CIBERER contracts. This work was funded by the Spanish Ministry of Science and Innovation (grant SAF2015-64869-R-FEDER), the Fundació la Marató TV3 (20132330), Research Contract with SIDRA Medicine (Qatar), CIBERER ACCI 2017-U731, and the Generalitat de Catalunya (grant SGR2009-1355). We thank INSTRUCT, part of the European Strategy Forum on Research Infrastructures (ESFRI) and the Hercules Foundation Flanders for their support to the Nanobody discovery. We thank Nele Buys for the technical assistance during Nanobody discovery. Generation of nanobody 74 was supported by INSTRUCT proposal 1176. We thank Nick Berrow (Core Facility of Protein Expression; IRB Barcelona) for providing us with HRV-3C protease, and Joan Pous from the IRB Barcelona/IBMB-CSIC crystallization platform. We also thank Marta Tauler from the Molecular Interaction Analysis Services (CCIT-UB) for Surface Plasmon Resonance Analysis of Nb74.

## Author contributions

E.E.-M. contributed in experimental design, acquisition, analysis, and interpretation of the data, writing the manuscript and approving it. J.F. contributed in experimental design, acquisition, analysis and interpretation of the data, revising the manuscript and approving it. P.B. contributed in experimental design, analysis and interpretation of the data, and approved it. L.D. contributed in experimental design, molecular dynamics experiments, interpretation of the data, revising and approving the manuscript. E.P. contributed in experimental design, nanobody generation, revising and approving the manuscript. X.C. contributed in experimental design, acquisition, analysis and interpretation of the data, revising and approving the manuscript. M.E.-G. contributed in revising and approving the manuscript. A.Z. contributed revising and approving the manuscript. C.Z. contributed in experimental design, acquisition, analysis and interpretation of the data, revising and approving the manuscript. J.S. contributed in experimental design, nanobody generation,

revising and approving the manuscript. J.F.-R. contributed in experimental design, molecular dynamics experiments, revising, and approving the manuscript. I.F. contributed in experimental design, acquisition, analysis and interpretation of the data, revising, and approving the manuscript. M.P. contributed in experimental design, writing the manuscript and approving it.

## Additional information

**Competing interests:** The authors declare no competing interests.

