## [Peer Review File · Nature Communications]

Reviewers' comments:

Reviewer #1 (Remarks to the Author):

The article by Errasti-Murugarren et al. reports the crystal structure of a bacterial homologue of the human L-type amino acid transporters LAT1 and hAsc-1 to 2.9Å resolution. Both LAT1 and hAsc-1 play important roles in maintaining amino acid homeostasis in normal cells, with several well known diseases resulting from their dysfunction and are up regulated in certain cancers, where they function to supply additional metabolites during uncontrolled cell division.

This structure thus represents an important advance in the field of amino acid transport. In particular the structure was determined in complex with a competitive inhibitor, 2-AIB, which the authors can observe sitting in the central ligand binding site. The authors support this observation, and subsequent mechanistic interpretations using both liposome based assays on specific variants of the bacterial transporter. A string point in this study is that the authors have then made equivalent mutations in the human Asc-1 transporter, supporting this homologue as a valid model system.

Overall I find this study well written and the conclusions well supported by the presented data. I have no major concerns and indeed, think this study can be accepted largely in the present form.

I do have several minor comments however, which the authors may wish to consider addressing in a revised version.

1. p9, line 259. Technically thermostability assays are not directly binding assays, so I would be careful not to conflate the two. For example, the current version assumes no change in K_d from this assay, which I would say is over interpretation of the data.
2. p9, line 267. I agree with the authors that their results suggest the difference observed in apparent K_d between extracellular and intracellular amino acid binding is a kinetic difference, which the authors explain mechanistically later in the paper. Indeed, a similar situation has already been reported for the plant nitrate transporter, NRT1.1, (Parker and Newstead, Nature 2014), where a change in apparent K_m is the result of changes in kinetics. It may be useful to discuss this in the paper, to support the authors claim that a similar mechanism is used in the SLC7 transporters.
3. Indeed, I was struck by the fact that the difference in amino acid 'binding' observed in both BasC and the mammalian LAT and Asc-1 transporters is essentially kinetic and not thermodynamic (as observed in NRT1.1). The authors may (or may not) wish to more explicitly spell this out in their article.
4. p29. Figure 1 - you may want to label your binding site in both panel a and c.
5. p32. Figure 4. The density shown in panels a and b does not look like 2.9Å, consider contouring differently?

Reviewer #2 (Remarks to the Author):

Comments

The authors solved the first crystal structure of a bacterial alanine-serine-cysteine antiporter, a member of the L-type amino acid (LAT) family whose human members are of pathophysiological relevance. This structure adopts a fold representative of the APC superfamily, the second largest family of secondary transporters. The authors have examined it to decipher the substrate binding and

get insight into the transport cycle.

I found this paper quite interesting although I have a few reservations (see hereafter) as I believe the authors could have gone deeper in the comparison with other antiporters and amino acid transporters of the APC superfamily.

1. As BasC exchanger is a homolog, albeit remote, of the proton-amino acid symporter ApcT and the amino acid antiporters AdiC and GadC, the authors should show a sequence alignment of BasC, with these proteins as well as with GkApcT. This would help when discussing the conservation/variability of amino acids at certain position and their role in the different transporters, if known. Although an alignment of BasC sequence with human LAT sequences is in the accompanying manuscript (Bartoccioni et al.) it would also be interesting to have it in the supplementary information.

2. The two BasC residues, Y236 and K154, are conserved in ApcT which is not an amino acid exchanger but a proton symporter. Similarly, Y236, but not K154, is conserved in AdiC which is an Arg/Agm antiporter which does not require (as the authors mentioned) the carboxylate moiety of the substrate. All these observations raise some doubts about the role proposed in this manuscript of Y236 and K154 as determinants of the asymmetry of the substrate affinity at the two sides of the membrane. The authors should comment on these points.

3. What is an α -amino carboxyl moiety? Do you mean a carboxylate moiety?

4. The authors claim that the nanobody does not induce a bias towards the inward-facing state based on a comparison with a nanobody-unbound BasC structure solved at 7.2 Å. However at this low resolution some structural differences cannot be easily pinpointed. In my opinion it is too strong a statement to write that: "The non-occluded inward-facing conformation structure of BasC in complex with 2-AIB fills a gap in the knowledge of the transport cycle of the APC transporter"

5. The authors should make a comparison of their inward-facing structures with other inward-facing APC structures (GkApcT, GadC, LeuT, ...)

Minor comments:

-Change "alanina" into "alanine" in Fig3b

Reviewer #3 (Remarks to the Author):

In this work by Errasti-Murugarren et al., the first crystal structures of bacterial Ala-Ser-Cys exchanger in complex with a nanobody in the apo form and in complex with the analogous substrate at moderate resolutions are presented. The structural study is technically sound, and the structural analysis has been performed well.

However the manuscript is written in such a way that it is difficult to comprehend, especially in the presentation of functional data. I had to reread it several times along with the submitted manuscript by Bartoccioni et al., to sort out what is what. There is still an outstanding issue that I do not understand - what is the explanation for a quite significant increase in the influx experiment (Fig.2a)? If this protein is an obligate exchanger, that should not be the case in my opinion.

Additionally the authors investigated the possible structural determinants of asymmetry in the affinities at extracellular and intracellular sites of the protein, and made a conclusion that conserved Y236 and K154 play the major role. In case of 154K the situation is very clear (and the authors nicely link the mutations at this residue to lysinuric protein intolerance in human homologue), however the

role of Y236 is less defined. The authors vaguely state that Y236F mutation (which is in fact the gain of function mutation, and the increase is not that small, about 20%) causes some alterations during the transport cycle but what exactly is meant by that? Does aromaticity play any role here? Would be nice to have a different mutant causing a reverse effect on transport. The same goes for S282A mutant. What is the role of S282 - can it be of any regulatory role? The mutation to something bulkier and /or causing stronger interaction with the carbonyl of G19 might give some additional insight. Out of curiosity - why did the authors move from radioactive L-Ser (used in Bartoccioni et al.,) to radioactive L-Ala (this study)?

For the rest, the manuscript needs to be brushed up - references should be in one style, in structural figures the unnecessary elements (such as main chain atoms shown as sticks and not interacting with the substrate or other atoms) should be removed to improve clarity, the type of electron density should be indicated (only in one place it is mentioned as POLDER map): 2fo-fc, fo-fc, omit
In the S1 table the values for completeness are different from those found in validation reports, I guess the reason is the anisotropic truncation; values of bond lengths rmsds, are off, I guess /hope it is 0.008 and not 0.8

The caption for S7 figure is very scarce, please describe each panel. Line 53: CA, aC, should be Ca
FigureS10, especially panel a, should be in the main text in my opinion
But certainly it is an important study, which will be of interest for a wide community - not only for structural biologists, but also pharmacologists, biochemists and other researchers.

Reviewers' comments:

Reviewer #1 (Remarks to the Author):

The article by Errasti-Murugarren et al. reports the crystal structure of a bacterial homologue of the human L-type amino acid transporters LAT1 and hAsc-1 to 2.9Å resolution. Both LAT1 and hAsc-1 play important roles in maintaining amino acid homeostasis in normal cells, with several well known diseases resulting from their dysfunction and are up regulated in certain cancers, where they function to supply additional metabolites during uncontrolled cell division.

This structure thus represents an important advance in the field of amino acid transport. In particular the structure was determined in complex with a competitive inhibitor, 2-AIB, which the authors can observe sitting in the central ligand binding site. The authors support this observation, and subsequent mechanistic interpretations using both liposome based assays on specific variants of the bacterial transporter. A string point in this study is that the authors have then made equivalent mutations in the human Asc-1 transporter, supporting this homologue as a valid model system.

Overall I find this study well written and the conclusions well supported by the presented data. I have no major concerns and indeed, think this study can be accepted largely in the present form.

I do have several minor comments however, which the authors may wish to consider addressing in a revised version.

1. p9, line 259. Technically thermostability assays are not directly binding assays, so I would be careful not to conflate the two. For example, the current version assumes no change in K_D from this assay, which I would say is over interpretation of the data.

The reviewer is completely right and we have changed “revealed” by “suggested” in the sentence interpreting thermostability results. Additionally, we have added “apparent” before K_D assuming that thermostability is not measuring directly binding and added that Tyr 236 **would** make a small contribution to substrate binding. Thanks for the observation. Text has been changed accordingly in lines 252-257.

2. p9, line 267. I agree with the authors that their results suggest the difference observed in apparent K_D between extracellular and intracellular amino acid binding is a kinetic difference, which the authors explain mechanistically later in the paper. Indeed, a similar situation has already been reported for the plant nitrate transporter, NRT1.1, (Parker and Newstead, Nature 2014), where a change in apparent K_m is the result of changes in kinetics. It may be useful to discuss this in the paper, to support the authors claim that a similar mechanism is used in the SLC7 transporters.

Great observation. Reference (Parker and Newstead, Nature 2014) has been added to the Discussion, and the text changed (lines 388-389).

3. Indeed, I was struck by the fact that the difference in amino acid 'binding' observed in both BasC and the mammalian LAT and Asc-1 transporters is essentially kinetic and not thermodynamic (as observed in NRT1.1). The authors may (or may not) wish to more explicitly spell this out in their article.

Another great point raised by the reviewer. Nevertheless, as is an issue that we are currently developing in our laboratory, we preferred not to go deeper on that, although we realized that is an important feature associated with the transporter mechanism.

4. p29. Figure 1 - you may want to label your binding site in both panel a and c.

Done. Binding site has been labelled in Figure 1 panel c as suggested by the reviewer. Nevertheless, as Figure 1a represents the Apo structure of BasC, so the substrate is not there, we preferred not to label it.

5. p32. Figure 4. The density shown in panels a and b does not look like 2.9A, consider contouring differently?

Reviewer is right. In fact electron density map qualities in this region look quite similar for the apo (at 2.9 Å Fig. 4a) and the holo (at 3.4 Å Fig. 4b) structures. Although this region is not representative of the overall electron density map quality, in particular for the apo structure, we prefer to keep Figure 4 because of its relevance as part of the proposed mechanism for the Y236F mutant.

Reviewer #2 (Remarks to the Author):

Comments

The authors solved the first crystal structure of a bacterial alanine-serine-cysteine antiporter, a member of the L-type amino acid (LAT) family whose human members are of pathophysiological relevance. This structure adopts a fold representative of the APC superfamily, the second largest family of secondary transporters. The authors have examined it to decipher the substrate binding and get insight into the transport cycle.

I found this paper quite interesting although I have a few reservations (see hereafter) as I believe the authors could have gone deeper in the comparison with other antiporters and amino acid transporters of the APC superfamily.

1. As BasC exchanger is a homolog, albeit remote, of the proton-amino acid symporter ApcT and the amino acid antiporters AdiC and GadC, the authors should show a sequence alignment of BasC, with these proteins as well as with GkApcT. This would help when discussing the

conservation/variability of amino acids at certain position and their role in the different transporters, if known. Although an alignment of BasC sequence with human LAT sequences is in the accompanying manuscript (Bartoccioni et al.) it would also be interesting to have it in the supplementary information.

BasC sequence alignment with human LATs and members of the APC family with known structures (AdiC, ApcT, GadC and GkApcT) has been added as Supplementary Figure 8, as requested by the reviewer.

2. The two BasC residues, Y236 and K154, are conserved in ApcT which is not an amino acid exchanger but a proton symporter. Similarly, Y236, but not K154, is conserved in AdiC which is an Arg/Agm antiporter which does not require (as the authors mentioned) the carboxylate moiety of the substrate. All these observations raise some doubts about the role proposed in this manuscript of Y236 and K154 as determinants of the asymmetry of the substrate affinity at the two sides of the membrane. The authors should comment on these points.

Great point that in our opinion helped to improve the discussion. Nevertheless, ApcT is an amino acid exchanger although not obligatory (as it is able to return empty when an acidic extraliposomal pH is present and independently of a pH gradient, i.e., acidic pH-activated non-obligatory exchanger). Unfortunately, there is not enough functional information on ApcT transporter to corroborate or to reject a similar role for both Lys 158 and Tyr 236 (ApcT numbering) on this transporter function. Nevertheless, what is known is that mutation of Lys 158 in ApcT (equivalent to Lys 154 in BasC) dramatically reduces alanine uptake, similarly to BasC (and GkApcT). This point has been added to Discussion section, pages 413-417.

In the case of AdiC, a complete functional characterization, including its sidedness for both arginine and agmatine, is available. In this regard, AdiC, that has also the conserved Tyr 236 (Tyr 239 AdiC numbering), is moderately asymmetric (in terms of apparent affinity; two fold higher intracellular K_m compared to the extracellular one) for both arginine and agmatine (which lacks the alpha-carboxyl group) (Reference 44). MD simulations and alanine binding results reported in the present paper for BasC, together with the observed moderate asymmetry for agmatine in AdiC, suggest that interaction of Tyr 236 with the substrate alpha-carboxyl group has nothing to do with this change in substrate apparent affinity. This is why we suggest that the differential interaction of BasC Tyr 236 with TM6 (via Ala 200) and TM1 (via Gly 19) would result in differential tension of the unwound regions which could affect translocation process, affecting in that way the catalytic constant and, thus, the intracellular substrate apparent affinity. In this regard, we are currently working to reveal the molecular bases underlying this effect. This has been discussed in lines 396-405.

3. What is an α -amino carboxyl moiety? Do you mean a carboxylate moiety?

Thanks for the observation. “ α -amino carboxyl moiety” has been changed to “ α -amino and α -carboxyl moieties” all along the document, as suggested by the reviewer.

4. The authors claim that the nanobody does not induce a bias towards the inward-facing state based on a comparison with a nanobody-unbound BasC structure solved at 7.2 Å. However at this low resolution some structural differences cannot be easily pinpointed. In my opinion it is too strong a statement to write that: “The non-occluded inward-facing conformation structure of BasC in complex with 2-AIB fills a gap in the knowledge of the transport cycle of the APC transporter”.

The reviewer is right stating that at 7.2 Å resolution some structural differences cannot be pinpointed. For this reason we have re-written the sentence to make it more acceptable suggesting that Nb74 does not select inward facing conformation (lines 154-156).

Regarding the sentence “The non-occluded inward-facing conformation structure of BasC in complex with 2-AIB fills a gap in the knowledge of the transport cycle of the APC transporter **family**” we must say that we refer (as stated at the end of the sentence) to the APC family, which includes only AdiC, ApcT, GkApcT and GadC. In this regard, our structure is, in fact, filling a gap as there is no other non-occluded inward-facing substrate-bound structure within this family. Indication that LATs and the solved AdiC, ApcT, GadC and GkApcT transporters belong to the APC family within the APC **superfamily** has been included in line 99.

5. The authors should make a comparison of their inward-facing structures with other inward-facing APC structures (GkApcT, GadC, LeuT, ...).

Good point raised by the reviewer. We have included a Supplementary Figure 5 where the structural superimposition of BasC with GadC and with other inward-facing transporters structures from the APC superfamily is shown. Although similar, particular differences can be observed, especially in the tilting of TMs1a, 5 and 7. This has been added to the Discussion section (lines 314-317). Additionally, results from structural superimposition of BasC with GadC (Supplementary Figure 5a) have been added to the Results section (lines 185-190).

Minor comments:

-Change “alanina” into “alanine” in Fig3b. Done as requested.

Reviewer #3 (Remarks to the Author):

In this work by Errasti-Murugarren et al., the first crystal structures of bacterial Ala-Ser-Cys exchanger in complex with a nanobody in the apo form and in complex with the analogous substrate at moderate resolutions are presented.

The structural study is technically sound, and the structural analysis has been performed well.

However the manuscript is written in such a way that it is difficult to comprehend, especially in the presentation of functional data. I had to reread it several times along with the submitted manuscript by Bartoccioni et al., to sort out what is what. There is still an outstanding issue that I do not understand - **what is the explanation for a quite significant increase in the influx experiment (Fig.2a)? If this protein is an obligate exchanger, that should not be the case in my opinion.**

The reviewer is right, but, in fact, the presented amino acid influx was not significantly different when compared in the presence and absence of Nb74. Indeed the tendency for Nb74 activation of BasC influx was mainly due to a unique experiment of the three experiments originally performed (the highest point in the Figure 1 for reviewer 3). Nevertheless we have performed a new set of experiments increasing the n to 6 independent experiments and the results are even clearer than before, confirming the lack of effect of Nb74 on alanine influx. To build the new Figure 2a in the manuscript, Grubbs's test to identify outliers eliminating the data indicated above [(Graphpad) alpha = 0.1], has been applied (<https://www.graphpad.com/guides/prism/7/statistics/index.htm>). Thanks for this observation that helped to clarify an important result in our manuscript.

Figure 1. Effect of Nb74 on L-Ala/L-Ala influx and efflux in BasC proteoliposomes. Representation with individual data from independent experiments with representation of the mean \pm s.e.m.

Additionally the authors investigated the possible structural determinants of asymmetry in the affinities at extracellular and intracellular sites of the protein, and made a conclusion that conserved Tyr 236 and Lys 154 play the major role. In case of Lys 154 the situation is very clear (and the authors nicely link the mutations at this residue to lysinuric protein intolerance in human homologue), however the role of Tyr 236 is less defined. **The authors vaguely state that Y236F mutation (which is in fact the gain of function mutation, and the increase is not that small, about 20%) causes some alterations during the transport cycle but what exactly is meant by that?**

The gain of function is a direct consequence of the increased intracellular apparent affinity for L-alanine. In that way, at 4 mM L-alanine (the intraliposomal L-alanine concentration used for kinetic characterization of BasC), about 80% of BasC wild-type is saturated, while Y236F mutant is fully saturated at this concentration (Figure 2 for reviewer 3). This gives an increase of about 20% in function when the exchange of extracellular 10 μ M [3 H]L-alanine is measured against 4 mM intraliposomal L-alanine (Figure 2 for reviewer 3).

Figure 2. Kinetic characterization of [3 H]L-alanine uptake in wild-type and Y236F mutant BasC proteoliposomes. Michaelis-Menten plot of the transporter-mediated uptake of [3 H]L-alanine (10 μ M, 1 μ Ci, 4 s) in either wild-type or Y236F mutant BasC-GFP proteoliposomes containing (0.1-6 mM) cold L-alanine. Mediated transport was calculated as [3 H]L-alanine uptake in L-alanine-containing PLs minus empty PLs. Data correspond to a representative experiment, performed using three replicates.

The molecular bases of this effect is still unknown (but we are currently working on that issue), as stated in the paper (lines 389-390). Nevertheless, we suggest, based on the MD and binding results that interaction of Tyr 236 with the substrate alpha-carboxyl group has nothing to do with this change in substrate apparent affinity. In this regard, AdiC, that has also the conserved Tyr 236, is moderately asymmetric (in terms of apparent affinity; two fold higher intracellular K_m compared to the extracellular one) for agmatine, which lacks the alpha-carboxyl group. All these observations suggested that is not the interaction with the substrate but another issue the one responsible for the low intracellular substrate apparent affinity. In this regard, we propose that the differential interaction of Tyr 236 with TM6 (via Ala 200) and TM1 (via Gly 19) would result in differential tension of the unwound regions which could affect translocation process and/or even substrate binding in the outward-facing conformation. As molecular bases underlying this mutant effects are under

current study, but still unknown, we have added a sentence in this sense in lines 403-405 of Discussion section.

Does aromaticity play any role here? Would be nice to have a different mutant causing a reverse effect on transport.

[Redacted]

Great point raised by the reviewer. In fact we have found some interesting issues regarding Tyr 236 aromaticity and its role in BasC function that are right now in progress.

[Redacted]

Nevertheless, we would prefer not to publish now these results as we have complementary observations that would help in the elucidation of the role of this residue on the LAT transporters translocation mechanism.

The same goes for S282A mutant. What is the role of S282 - can it be of any regulatory role? The mutation to something bulkier and /or causing stronger interaction with the carbonyl of G19 might give some additional insight.

Great observation. We think S282 has a regulatory role of the inward to outward transition affecting in that way transporter Vmax. In fact, S282A mutant has a small increase on Vmax but not on apparent affinity (attached Figure 4a for reviewer 3). Substitution of Ser 282 by a cysteine greatly reduces L-alanine uptake (Figure 4b for reviewer 3). In fact introduction of any residue bigger than a serine results in steric restrictions between residue in position 282 and TM1 (Figure 4c for reviewer 3). In this regard all the LAT transporters as well as the crystallized APC family members have a serine or glycine residue in the equivalent position to S282 (Supplementary Figure 8).

Figure 4. Functional and structural analysis of Ser 283 BasC mutants. **a)** Michaelis-Menten plot of the transporter-mediated uptake of [³H]L-alanine (1 μCi, 4 s) in either wild-type or S282A mutant BasC-GFP proteoliposomes containing 4 mM L-alanine, varying extracellular L-alanine concentrations (0-250 μM). Mediated transport was calculated as [³H]L-alanine uptake in L-alanine-containing PLs minus empty PLs. Data correspond to a representative experiment, performed using three replicates. **b)** 10 μM [³H]L-alanine (1 μCi/μl, 4 s) influx (pmol/μg protein · s) into cysless (C427S) and S282C (in cysless background) BasC-GFP proteoliposomes containing 4 mM L-alanine. Transport was expressed as the percentage of [³H]L-alanine transport, considering 100% [³H]L-alanine uptake by the cysless BasC protein. **c)** Mutations of S282 to cysteine or to asparagine generates clashes with residue Gly 19 in the BasC structure bound to 2-AIB. Data correspond to mean±s.e.m. of three independent experiments.

As is the case of the role of Tyr236 aromaticity on BasC function, we are also studying the role of S282 on BasC function as we think that interaction of TM1 and TM8 via S282-G19 regulates inward to outward transition velocity. Again,

we would prefer not to publish now these results as they are part of an ongoing project.

Out of curiosity - why did the authors move from radioactive L-Ser (used in Bartoccioni et al.,) to radioactive L-Ala (this study)?.

It was a matter of simplicity, as reviewers of the Bartoccioni et al paper had some difficulties to understand the use of two different substrates. Additionally, alanine/alanine exchange is even slower than alanine/serine exchange (so being lineal for longer periods of time), which allowed us to functionally characterize BasC mutants more precisely.

For the rest, the manuscript needs to be brushed up - **references should be in one style**, Done as requested.

in structural figures the unnecessary elements (such as main chain atoms shown as sticks and not interacting with the substrate or other atoms) should be removed to improve clarity,

Done as requested by the reviewer. In Figure 3 residues Gly 203, Ile 18 and Gly 19 that are not interacting neither with the substrate nor with other atoms have been eliminated. In the rest of the figures we have decided to maintain all atoms as electronic densities are shown.

the type of electron density should be indicated (only in one place it is mentioned as POLDER map): 2fo- ρ , fo- ρ , omit.

Information has been added to Figures legends for Figure 4a, 5a and Supplementary Figure 3, as requested by the reviewer.

In the S1 table the values for completeness are different from those found in validation reports, I guess the reason is the anisotropic truncation; values of bond lengths rmsds, are off, I guess /hope it is 0.008 and not 0.8.

Reviewer is totally right. Discrepancies are due to changes in the X-ray data before and after treatment of "raw/processed" data with the anisotropy server from UCLA for the anisotropic correction, as stated in Materials and Methods section. We think this is now part of a general important topic of discussion for data deposition at the PDB as reflected in the very recent advise/report from Dr. G. Bricogne (http://staraniso.globalphasing.org/deposition_about.html) to the X-ray community, although a consensus is not yet fully settled. To try to make things (a bit) clearer we have added an extra row to Supplementary Table 1 indicating completeness both before and after the anisotropic correction (which was the data used during refinement and deposited at the PDB). Thanks to the reviewer for this observation.

Regarding bond lengths and angles, in the original Table we included RMSZ (Root Mean Squared Z) score for both bond length and angles from the

validation report. As suggested by reviewer we changed to the commonly used RMSD values.

The caption for S7 figure is very scarce, please describe each panel. Line 53: CA, α C, should be C α .

Legend for Supplementary Figure 7 has been updated and completed as requested by the reviewer, including CA, which has been defined in the legend.

FigureS10, especially panel a, should be in the main text in my opinión.

Supplementary Figure 10 has been added to the main text as Figure 7 as requested by the reviewer.

But certainly it is an important study, which will be of interest for a wide community - not only for structural biologists, but also pharmacologists, biochemists and other researchers.

Thanks a lot for the observation.

REVIEWERS' COMMENTS:

Reviewer #2 (Remarks to the Author):

I find the revised version of this manuscript now acceptable for publication. I still have two minor remarks (see below).

1. In the Introduction pg. 3 line 95:

To the best of my knowledge, AdiC and GadC are not amino acid exchangers. AdiC and GadC indeed import amino acids to the cytosol but export the decarboxylated forms of these amino acids.

2. In the added paragraph of the Discussion section (pg. 13):

"In this regard, Tyr 236, which is conserved in AdiC (Supplementary Figure 8), could play a role

I would write :

"In this regard, Tyr 239, the corresponding residue in AdiC of BasC Tyr 236, (Supplementary Figure 8), could play a role"

And

"In fact, the reported asymmetry for agmatine, which lacks the α -carboxyl group, reinforces the idea that interaction of Tyr 236 hydroxyl group with substrate α -carboxyl is not essential for..."

I would write:

"In fact, the reported asymmetry for agmatine, which lacks the α -carboxyl group, reinforces the idea that, similarly to Tyr 236 in BasC, interaction of Tyr 239 hydroxyl group with substrate α -carboxyl is not essential"

3. pg. 13 line 407: Change "substrate K_D " into "substrate K_D "

Reviewer #3 (Remarks to the Author):

I would like to thank authors for very clear response to the raised concerns. I believe the manuscript is in a good shape now to be published.

Point by point answer to reviewer's points

Manuscript entitled "L-Amino acid Transporter structure and molecular bases for the asymmetry of substrate interaction" by Ekaitz Errasti-Murugarren, Joana Fort, Paola Bartoccioni, Lucía Díaz, Els Pardon, Xavier Carpena, Meritxell Espino-Guarch, Antonio Zorzano, Christine Ziegler, Jan Steyaert, Juan Fernández-Recio, Ignacio Fita and Manuel Palacín

REVIEWERS' COMMENTS:

Reviewer #2 (Remarks to the Author):

I find the revised version of this manuscript now acceptable for publication. I still have two minor remarks (see below).

1. In the Introduction pg. 3 line 95: To the best of my knowledge, AdiC and GadC are not amino acid exchangers. AdiC and GadC indeed import amino acids to the cytosol but export the decarboxylated forms of these amino acids. Sentence changed as indicated by the reviewer. We used the word transporter to avoid the need to specify that they exchange amino acids and their decarboxylated forms.

2. In the added paragraph of the Discussion section (pg. 13):

"In this regard, Tyr 236, which is conserved in AdiC (Supplementary Figure 8), could play a role

I would write :

"In this regard, Tyr 239, the corresponding residue in AdiC of BasC Tyr 236, (Supplementary Figure 8), could play a role" Sentence changed as indicated by the reviewer.

And

"In fact, the reported asymmetry for agmatine, which lacks the α -carboxyl group, reinforces the idea that interaction of Tyr 236 hydroxyl group with substrate α -carboxyl is not essential for..."

I would write:

"In fact, the reported asymmetry for agmatine, which lacks the α -carboxyl group, reinforces the idea that, similarly to Tyr 236 in BasC, interaction of Tyr 239 hydroxyl group with substrate α -carboxyl is not essential" Sentence changed as indicated by the reviewer.

3. pg. 13 line 407: Change "substrate KDs " into "substrate KD". Done as indicated by the reviewer.

Reviewer #3 (Remarks to the Author):

I would like to thank authors for very clear response to the raised concerns. I believe the manuscript is in a good shape now to be published. Thanks; no action required.